# Evaluating Explanatory Evaluations: An Explanatory Virtues Framework for Mechanistic Interpretability

## Abstract

Mechanistic Interpretability (MI) aims to understand neural networks through causal explanations. Though MI has many explanation-generating methods and associated evaluation metrics, progress has been limited by the lack of a universal approach to evaluating explanatory methods. Here we analyse the fundamental question "What makes a good explanation?" We introduce a pluralist *Explanatory Virtues Framework* drawing on four perspectives from the Philosophy of Science—the Bayesian, Kuhnian, Deutschian, and Nomological—to systematically evaluate and improve explanations in MI. We find that Compact Proofs consider many explanatory virtues and are hence a promising approach. Fruitful research directions implied by our framework include (1) clearly defining explanatory **simplicity**, (2) focusing on **unifying** explanations and (3) deriving **universal principles** for neural networks. Improved MI methods enhance our ability to monitor, predict, and steer AI systems.

## 1 Introduction

Mechanistic Interpretability (MI) is the study of producing causal, scientific explanations of artificial neural networks (Bereska & Gavves, 2024; Sharkey et al., 2025; Olah et al., 2020). Good explanations allow us to monitor and understand AI systems as well as providing affordances for steering and debugging. But what is a *good explanation*? And how do we know that our methods for producing and evaluating explanations are effective at producing *good explanations*?

Wu et al. (2024) observe the following problem: When analysing the same algorithmic task, Chughtai et al. (2023) and Stander et al. (2024) produced what appeared to be two valid MI explanations of the same model. Yet the mechanisms that they propose are mutually inconsistent. Without systematic criteria for choosing between explanations, it is difficult to give good epistemic reasons for declaring one explanation to be the better one. Without good reasons to choose, researchers may either suspend judgement or resort to disparate and subjective preferences. [1]

Explanatory Methods typically come with two core components: Firstly, they have a *generative* component which produce explanations of model internals. Secondly, they have an *discriminative* component which evaluates the quality of the explanation and can be used to compare different explanations of the same type against each other. For example, the Sparse Autoencoder (SAE) method (Bricken et al., 2023; Ayonrinde et al., 2024; Huben et al., 2024; Gao et al., 2024) have a generative component the SAE model and accompanying (auto or human) semantic interpretability scheme (Bills et al., 2023; Paulo et al., 2024). SAEs also come with a discriminative component that states that explanations are higher quality if they Pareto dominate another explanation on the (accuracy, simplicity)-frontier.

Recent work has developed evaluation metrics for interpretability with respect to either specific methods (Karvonen et al., 2024), or specific synthetic tasks (Gupta et al., 2024; Thurnherr & Scheurer,

---

[1] Note that *faithfulness* here refers to *explanatory faithfulness* (Ayonrinde & Jaburi, 2025), explanations which match the step-by-step process of the model's computation, and not *behavioural faithfulness*, explanations that provide the same outputs as the original model when given the same input but plausibly using different algorithms.

2024). However, there is not a unifying framework that allows us to compare different explanatory methods across a wide variety of tasks.

To address this problem, we introduce the **Explanatory Virtues Framework**, which answers the question: *Given two competing explanatory theories, which should we prefer?* In particular, our framework provides a systematic way to an analyse explanatory methods and evaluations, where evaluations that do not (even at least implicitly) prefer explanations which embody the Explanatory Virtues are unlikely to produce ideal explanations. Our framework draws from the Philosophy of Science, specifically the *Bayesian*, *Kuhnian*, *Deutschian*, *Nomological* accounts of explanation and we apply their criteria for theory choice to MI methods. We examine the qualities that we should, and do, seek in good explanations, via theoretical analysis and case studies respectively. Using our Explanatory Virtues Framework, we analyse four Mechanistic Interpretability methods: Clustering, Sparse Autoencoders (SAEs), Causal Circuit Analysis, and Compact Proofs. We find that the following Explanatory Virtues are often neglected among current MI methods: *Simplicity*, *Unification*, *Co-Explanation*, and *Nomological Principles*. We hence suggest pursuing these virtues as promising research directions.

The task of choosing between explanations on the algorithmic task in Wu et al. (2024) drove them to use the *Compact Proofs* evaluation (Section 4.2). We evaluate the Compact Proofs evaluation approach and find that this approach embodies many of the Explanatory Virtues and is an effective means of determining which explanations should be preferred. Wu et al. (2024) demonstrated our framework's utility by applying the Compact Proofs methodology to three competing explanations: two prior explanations and their own. They found that the two previous interpretations failed to produce non-vacuous bounds (indicating poor Accuracy and Simplicity), while their interpretation succeeded. This exemplifies how our framework can resolve explanatory conflict.

The Explanatory Virtues Framework provides a systematic approach for evaluating MI methods and increasing our understanding of AI systems. Such understanding is useful for AI Safety, AI Ethics, and AI Cognitive Science (Bengio et al., 2025; Anwar et al., 2024; Chalmers, 2025), as well as debugging and improving neural networks (Lindsay & Bau, 2023; Sharkey et al., 2025; Amodei, 2025).

**Contributions.** Our contributions are as follows:

- Firstly, we provide a unified account of the Explanatory Virtues in MI. This can be understood as an answer to the question "What makes a good explanation?".
- Secondly, we analyse and compare MI methods with respect to these virtues.
- Finally, we suggest new directions for developing MI explanations, beyond the current state of the art.

## 2 VALID EXPLANATIONS IN MECHANISTIC INTERPRETABILITY

*Neural network interpretability* (henceforth just *interpretability*) is the process of understanding artificial neural networks using the scientific method. In this paper we focus on *Mechanistic Interpretability (MI)*. Following Ayonrinde & Jaburi (2025), we distinguish Mechanistic Interpretability from other forms of interpretability noting that Mechanistic Interpretability produces Model-level, Ontic, Causal-Mechanistic, and Falsifiable explanations.

### 2.1 EXPLANATIONS IN MECHANISTIC INTERPRETABILITY

Good scientific explanations provide answers to *why* questions. Typically a scientific explanation will provide an answer to the question "Why did the phenomenon occur?" and a good explanation will enable the listener to better comprehend the phenomenon. Explanations aim at knowledge. As compression and comprehension are closely linked (Wilkenfeld, 2019), good explanations *compress observations by exploiting regularities in data*.

Neural networks are classically viewed as black-box prediction machines (Lipton, 2018). However, Ayonrinde & Jaburi (2025) describe an alternative *Explanatory View* of Neural Networks, emphasising that deep neural networks contain *representations* and *mechanisms* that can be understood as providing implicit explanations for their behaviour. As models learn to generalize, they develop internal

structures that compress information about the world (Lehalleur et al., 2025). Good explanations uncover these internal structures.

## 2.2 Defining Mechanistic Interpretability

Following Olah et al. (2020), Ayonrinde & Jaburi (2025) define Mechanistic Interpretability as follows[2]:

> **Technical Definition of Mechanistic Interpretability (Ayonrinde & Jaburi, 2025)**
>
> Interpretability explanations are **valid** Mechanistic Interpretability explanations if they are **Model-level**, **Ontic**, **Causal-Mechanistic**, and **Falsifiable**.
>
> - **Model-level**: Explanations should focus on understanding the neural network and not the sampling method or other system-level properties (Arditi, 2024; Zaharia et al., 2024).
> - **Ontic**: Explanations should refer to real entities within the model (Salmon, 1984).
> - **Falsifiable**: Explanations should yield testable predictions (Popper, 1935).
> - **Causal-Mechanistic**: Explanations should identify a step by step continuous causal chain from cause to phenomena, rather than statistical correlations or general laws (Woodward, 2003; Salmon, 1989; Bechtel & Abrahamsen, 2005).

## 3 The Virtues of Good Explanations

"*Given two competing explanatory theories, which should we prefer?*" This is the question of *Theory Choice* (Kuhn, 1981; Schindler, 2018; Kuhn, 1962). To answer this question we may look at the properties of explanations. We refer to the truth-conducive properties of explanations as **Explanatory Virtues**. Explanatory Virtues are properties that are reliable indicators of truth.

Whether a property is an Explanatory Virtue is a *normatively* loaded; we should epistemically prefer explanations which embody Explanatory Virtues as such explanations are more likely to be true and the aim of scientific explanation is to aim at truth.[3] Conversely, we *descriptively* refer to properties of explanations that scientists value in practise as **Explanatory Values**.

In this section, we discuss Explanatory Virtues — the properties that ML researchers *should* value. We assess four accounts of explanation: the Kuhnian, Bayesian, Deutschian, and Nomological accounts. If these accounts correctly identify properties that we ought to value, then the combined set of such properties are Explanatory Virtues. These properties will form our pluralist **Explanatory Virtues Framework**. We provide a mathematical definition for each Explanatory Virtue which serves to ensure that there is a consistent and canonical way to compute each virtue thus allowing for a more objective comparison of explanations. Then in Section 4, we will discuss what MI researchers *do* value in practise, that is the Explanatory Values in Mechanistic Interpretability. We provide a summary of our Pluralist Explanatory Virtues Framework and how the virtues relate to each other in Figure 1.

**Notation.** We denote the explanation under consideration as $E \in \mathcal{E}$, where $\mathcal{E}$ is the set of all possible explanations and $B$, the background theory. $\mathbf{x_T}$ denotes observational data that the explanation is fitted to (training data). We assume $\mathbf{x_T}$ is sampled from the set of possible observational data $\mathcal{X}$. $\mathbf{x_I}$ denotes future observational data that was not accessible at explanation-making time (inference-time data). $x_{T,i}$ is the $i$-th data point in $\mathbf{x_T}$, where bolded $\mathbf{x}$ denotes a sequence of data points. We denote $k$ a complexity measure (for example, Kolmogorov complexity) and $|E|_B$ the description length of an explanation $E$ under background theory $B$ measured in bits.

---

[2] See Ayonrinde & Jaburi (2025) for a more complete exposition. Also see Appendix E.1 for intuitive examples of Explanation Types.

[3] Schindler (2018) provides a discussion of the truth-conduciveness of the virtues we discuss.

## 3.1 BAYESIAN THEORETICAL VIRTUES

Wojtowicz & DeDeo (2020) describe a Bayesian approach to Inference to the Best Explanation (Henderson, 2014). Here, the Explanatory Virtues are the credence-raising properties of the theory. These virtues can be split into two categories: **theoretical virtues** (in blue), which are properties of the explanation that do not depend on any observed or yet to be observed data, and **empirical virtues** (in orange), which are properties of the explanation that are defined in relation to the observed data.

**Accuracy, Precision, and Priors.** The Bayesian virtues are the empirical Explanatory Virtue of **Accuracy**, the theoretical Explanatory Virtue of **Precision** and the **Prior** probability of some explanation given the background theory.

Accuracy represents the probability of the true data given the explanation. Log-likelihood is the logarithm of Accuracy. Similarly, Precision is the expected log-likelihood of data conditional on the explanation being true. Precision represents the degree to which an explanation's predictions concentrate in a particular region of the space of possible observed data. Higher precision means that the explanation is more constraining in its predictions, making risky and useful predictions that rule out other possibilities, if the explanation is correct. [4]

We decompose Accuracy and Precision into further Explanatory Virtues as follows.

**Descriptiveness and Co-Explanation.** Given many data points $\mathbf{x} = \{x_1, x_2, \ldots, x_n\}$, we would like to understand how well an explanation explains each data point in isolation and how well it explains multiple data points together. We hence define **Descriptiveness** as the component of Log-Likelihood where data observation is considered in isolation and **Co-Explanation** as the component of Log-Likelihood which focuses on how an explanation can explain multiple data points, above its ability to predict any single observation in isolation.

**Power and Unification.** Analogously, we can break down Precision into our theoretical virtues of **Power** and **Unification**, defined analogously where Power measures the ability to explain individual data points and Unification measures the ability to connect multiple disparate observations together.

---

**Glossary of Bayesian Virtues**

$$Acc(E) = \mathbb{P}(\mathbf{x}_T | E) \tag{Accuracy}$$

$$Prec(E) = \mathbb{E}_{x_T \sim \mathcal{X}}[\log(\mathbb{P}(\mathbf{x}_T | E))] \tag{Precision}$$

$$Prior(E) = \mathbb{P}(E | B) \tag{Prior}$$

$$Desc(E) = \sum_i \log(\mathbb{P}(x_{T,i} | E)) \tag{Descriptiveness}$$

$$CoEx(E) = \log(Acc(E)) - Desc(E) = \log\left(\frac{\mathbb{P}(\mathbf{x}_T | E)}{\prod_i \mathbb{P}(x_{T,i} | E)}\right) \tag{Co-Explanation}$$

$$Power(E) = \mathbb{E}_{x_T \sim \mathcal{X}}\left[\sum_i \log(\mathbb{P}(x_{T,i} | E))\right] \tag{Power}$$

$$Unif(E) = Prec(E) - Power(E) = \mathbb{E}_{x_T \sim \mathcal{X}} \log\left(\frac{\mathbb{P}(\mathbf{x}_T | E)}{\prod_i \mathbb{P}(x_{T,i} | E)}\right) \tag{Unification}$$

---

[4]Note that the definition of Precision here is a slightly different notion to the Precision metric in Machine Learning as in 'Precision-Recall' analysis (Hastie et al., 2009). There, Precision is the fraction of true positives among the predicted positives. Here, by Precision we mean to say that more precise explanations are more constraining in their predictions.

## 3.2 KUHNIAN THEORETICAL VIRTUES

Kuhn (1981) lists five theoretical virtues as a basis for theory choice: **Accuracy**, **(Internal) Consistency**, **Scope (Unification)**, **Simplicity** and **Fruitfulness**. We previously explored Unification (Scope) and Accuracy in Section 3.1.

**Accuracy and Fruitfulness.** Accuracy is the extent to which the explanation fits the available data at the time of the creation of such an explanation. We can think of this as the "mundane empirical success" of an explanation, which we can contrast with the "novel empirical success" of an explanation or its **Fruitfulness** (Lakatos, 1978). Machine Learning researchers may draw a close analogy here with Accuracy being a performance measure on the training/validation set and Fruitfulness being a performance measure on a (naturally held-out) test set. Fruitful explanations have reach: they usefully generalise beyond the context of the original problem that the explanation was designed to solve.

**Consistency.** A necessary criterion for a theory to be a good explanation is that it is internally consistent. That is to say, the explanation must not contain any logical contradictions.

**Simplicity.** *Simplicity* is considered a key virtue for scientific explanations (White, 2005; Qu, 2023; MacKay, 2003). However, there are many forms of simplicity that may be chosen, which may rank explanations differently (Lakatos, 1970). We consider the main three forms of measures of simplicity: *Parsimony*, *Conciseness* and *Complexity*. **Parsimony** counts the number of entities that are posited by the explanation (Wojtowicz & DeDeo, 2020).[5] **Conciseness** is a Shannon-complexity measure of the information in an explanation given by the description length (Shannon, 1948; MacKay, 2003), **(K-)Complexity** is a Kolmogorov-complexity measure of an explanation in terms of the shortest program that can generate it (Kolmogorov, 1965; Hutter et al., 2024). For all simplicity measures, lower values are preferred.

---

**Glossary of Kuhnian Virtues**

$$
\begin{aligned}
Fruit(E) &= \quad \mathbb{P}(\mathbf{x}_I | E) &&\text{(Fruitfulness)} \\
\text{E is inconsistent} &\iff E \vDash \bot &&\text{(Consistency)} \\
Pars(E) &= \quad \#\_of\_entities(E) &&\text{(Parsimony)} \\
DL(E) &= \quad |E|_B &&\text{(Conciseness)} \\
\text{k-Compl}(E) &= \quad k(E) &&\text{(Complexity)}
\end{aligned}
$$

---

## 3.3 DEUTSCHIAN THEORETICAL VIRTUES

**Falsifiability and Hard-to-Varyness.** Popper (1935) writes that the key criteria of science is that its theories should be **Falsifiable** - that is, our explanations should come with a clear set of testable predictions attached. Deutsch (2011) further argues that alongside falsifiability, we should also seek explanations which themselves are **Hard-To-Vary**. Intuitively we might think of an explanation E as hard-to-vary if it cannot be easily modified to account for incoming data that contradicts the explanation. More precisely consider a modification $\Delta$ to an explanation E, where $\Delta$ is some edit operation formed of a list of insertions, deletions, substitutions and transpositions of symbols in E. $|\Delta|$ is the number of such operations in $\Delta$.

The hard-to-varyness criteria then captures the intuition that if you add some modification or "epicycle" $\Delta$ to an explanation E, then the new explanation E' should have lower novel empirical success than E (complexity-weighted). Conversely, if we can add some modification to an explanation and the new

---

[5] Parsimony is slippery to define well in practise as it is not always clear what counts as an entity. Worse still, parsimony might treat intuitively highly complex objects and very simple objects both equivalently as "entities" and simply count them up without nuance. Baker (2022) provides a discussion of the downsides of Parsimony as a measure of simplicity.

explanation has higher mundane and novel empirical success without being more complex, then we should prefer the new explanation.[6]

For some complexity measure k, we can then say that an explanation E is hard-to-vary if it is at a local maximum of the function $hv(E) = \log(Acc(E)) - k(E)$.[7]

> **Hard-to-Varyness**
>
> An explanation $E$ is hard-to-vary if it is at a local maximum of the function
> $$hv(E) = \log(Acc(E)) - k(E) \qquad \text{(Hard-to-Varyness)}$$

### 3.4 NOMOLOGICAL THEORETICAL VIRTUES

In Hempel & Oppenheim (1948)'s Deductive-Nomological (DN) model of explanation, a scientific explanation is a *sound deductive* argument where at least one of the premises is a "general law". For our purposes, we can think of general laws as "for all" statements which are true and not accidentally true. General laws describe necessary rather than contingent facts of the world. For example, "all gases expand when heated under constant pressure" is a general law whereas "all members of the Greensbury School Board for 1964 are bald" might be true but only by coincidence, as it were.

**Nomologicity.** Though we do not require our explanations to precisely follow the DN model of explanation, the **Nomologicity** (or *Lawfulness*) of an explanation, i.e. whether the explanation appeals to general laws or derives universal principles, is an explanatory virtue.

> **Nomologicity**
>
> An explanation $E$ is nomological if it appeals to general laws or universal principles about neural networks.

## 4 EXPLANATIONS IN THE WILD: CASE STUDIES IN MECHANISTIC INTERPRETABILITY

In Section 3, we explored the Explanatory Virtues. These values included the Theoretical Explanatory Virtues of *Precision*, *Power*, *Unification*, *Consistency*, *Simplicity*, *Nomologicity*, *Falsifiability* and *Hard-To-Varyness* as well as the Empirical Explanatory Virtues of (Mundane) *Accuracy*, *Descriptiveness*, *Co-Explanation* and *Fruitfulness*. We now consider how these virtues are instantiated in the methods that Mechanistic Interpretability researchers use in practice. That is, we consider how *valued* each Explanatory Virtue is within MI methods.

We note that we are not evaluating particular explanations (that may be produced from MI methods) and asking whether this explanation scores highly on some property (e.g. accuracy or simplicity) but are instead evaluating whether the explanatory method values a given virtue at all. We provide a rubric for evaluating whether an explanatory method embodies a virtue in Table 2. Visual summaries of the methods we discuss in this section can be found in Appendix D.

### 4.1 EXAMPLES

#### 4.1.1 CLUSTERING (ACTIVATIONS OR INPUTS)

One primitive form of neural network explanation is a clustering of model inputs or activations. For a complex model, such an explanation will not typically be highly accurate. However, this explanation *is* a simplification of the overall model performance. Here we might imagine finding some partition of the input/activation space, mapping a given input $\mathbf{x}$ to its associate cluster, of which $\mathbf{x}$ is ideally a

---

[6] We provide a complementary adhocness metric in Appendix F.

[7] We informally consider two explanations close if they are a small number of edit operations apart.

typical member. Then we may take the cluster (and possibly the output of the model on some cluster representative) as a proxy for the model's behaviour. [8]

Though this explanation is clearly not sufficient in many cases, we note that it does perform some compression of the input space and we can control the simplicity of the explanation by varying the number of clusters. Similarly, the explanation generated here is Falsifiable; we can test how well our cluster model predicts the behaviour of the original model. However, this explanation clearly falls down by not being Causal-Mechanistic in nature, and the Fruitfulness of the explanation may be low if the procedure is vulnerable to outliers.

### 4.1.2 SPARSE AUTOENCODER EXPLANATIONS OF REPRESENTATIONS/ACTIVATIONS

Sparse Autoencoders (SAEs) can be used to decompose the representations of neural activations into a linear combination of sparsely activating, disentangled and monosemantic latents (Bricken et al., 2023; Huben et al., 2024). Though many evaluation schemes have been proposed for SAEs (Karvonen et al., 2024; Wu et al., 2025), the primary axes on which SAE explanations are evaluated is on *Empirical accuracy* and *Simplicity*. Here Accuracy represents either a local unsupervised accuracy measure like reconstruction error, or the downstream performance of the interpreted model when the SAE reconstructions are patched into the model in place of the original activations.

**MDL-SAEs.**  Ayonrinde et al. (2024) provide a useful case study of how different types of Simplicity measures may be more or less principled in different contexts. Within the MDL-SAE (Minimum Description Length SAE) framework, SAE explanations are evaluated on Accuracy, Novel Empirical Success and Conciseness, where *Conciseness* is an information theoretic measure of Simplicity (see Section 3.2). This stands in contrast to the classical SAE framework where the simplicity measure is instead the SAE latent sparsity, a *parsimony* measure. In this case, changing the simplicity measure from sparsity (Parsimony) to description length (Conciseness) solved three key problems for SAEs: avoiding undesired feature splitting, enabling principled choice of SAE width, and ensuring uniqueness of feature-based explanation (Ayonrinde, 2024).

**EVF for SAEs.**  SAE explanations, like most ML methods, value Falsifiability and Novel Empirical Success (predictions beyond the training set). There is also some Co-Explanatory power in that the same feature dictionary should be used to explain any activations (at least from the same layer of the model). However, SAE explanations might be Ad-hoc and not Hard-to-Vary. As noted by Braun et al. (2024), contributions from features activated on SAEs trained for reconstruction may have little effect on the downstream performance of the model. Hence the corresponding feature activations are effectively free parameters. Similarly, the tendency to enlarge the feature dictionary (i.e. increase the SAE width) or add additional active features to explanations (i.e. increase the allowable $\ell^0$ norm of the feature activations vector) without clear justification, suggests an implicit ad-hocness in the explanations. MDL-SAEs provide some guidance against the ever increasing size of the feature dictionary, however it still remains an open question as to how to ensure that SAE explanations are truly hard-to-vary and pick out features which are causally relevant to the downstream behaviour of the model (Leask et al., 2025).

### 4.1.3 CAUSAL ABSTRACTION EXPLANATIONS OF CIRCUITS

As in neuroscience, a natural way to explain the behaviour of a neural network for interpretability researchers is to decompose the network into circuits (Olah et al., 2020; Kandel et al., 2000). Circuits can be formally specified by a correspondence between the network and some understood high-level causal model using the theory of Causal Abstractions (Geiger et al., 2023; Woodward, 2003; Beckers & Halpern, 2019; Pearl, 2009). In particular, the notion of abstraction that is typically appealed to is constructive abstraction (Beckers & Halpern, 2019). Paraphrasing from Geiger et al. (2021), a high-level model (an understandable causal model) is a *constructive abstraction* of a low-level model if we can partition the variables in the low-level model (e.g. the neural network neurons) such that:

1. Each low-level partition cell can be assigned to a high-level variable.

---

[8] We may think of the clustering explanation as performing some "quotienting" operation of the input space by the equivalence relation of being in the same cluster.

2. There is a systematic correspondence between interventions on the low-level partition cells and interventions on the high-level variables.

The Causal Abstraction framework for circuit analysis clearly focuses on the Falsifiability of explanations and the *Faithfulness* of the explanation to the underlying causal model (Empirical Accuracy and Novel Success under interventions). To encourage simplicity in explanations, we may also seek *Completeness* and *Minimality* in circuit explanations (Wang et al., 2023). (Behavioural) Faithfulness, Completeness, and Minimality are denoted the *FCM* criteria for circuit explanations (see Appendix K)

Algorithms such as ACDC (Conmy et al., 2023) find circuits that (approximately) satisfy the FCM criteria. However, it is well known (Wang et al., 2023) that the FCM criteria are in tension and that it is not always possible to satisfy all three criteria simultaneously. In practise, finding circuits is a computationally challenging problem and circuit discovery algorithms typically only find approximately optimal circuits (Adolfi et al., 2024).

**EVF for Circuit Explanations.** Despite the virtues of these approaches, they however do suffer from poor unification, co-explanation and nomologicity. In both manual and automated circuit discovery methods, most attention is paid to individual circuits rather than the relation and composition of subcircuits. Circuit explanations for two related tasks which share internal components are not typically privileged. Similarly, there are often no general laws or principles that detail which circuits are likely to be found in a network, and how these circuits relate to one another across contexts.

## 4.2 COMPACT PROOFS

The above examples of Clustering, SAEs and Circuits are methods for both the *creation* of explanations and also provide *evaluation methods* for the explanations created. The Compact Proofs methodology (Gross et al., 2024; Wu et al., 2024; Jaburi et al., 2025) is a method for evaluating *any* Causal-Mechanistic explanations obtained through other methods. In the Compact Proofs framework, an explanation is converted into a formal guarantee that allows researchers to assess the Accuracy and Simplicity of the explanation. We refer to Appendix J for a glossary of terms used in this section.

Given a data distribution $\mathcal{D}$, and a model $M_\theta$ with weights $\theta \in \mathcal{W}$, we would like to obtain a lower bound for the model's accuracy over $\mathcal{D}$.[9] Formally, we construct a *verifier* program $V(\theta, E)$, where $E$ is the explanation. The aim for $V$ is to return a bound on the model's performance that is as tight as possible whilst requiring that the proof of that bound that is as computationally efficient as possible. We may think of the computational efficiency as a measure of the simplicity of the proof (Xu et al., 2020). Note that these two goals, the *tightness* (Accuracy) of the bound and the *compactness* (Simplicity) of the proof (explanation), are in tension with one another. A good explanation should push out the (tightness, compactness)-Pareto frontier.[10]

Gross et al. (2024) show that faithful mechanistic explanations lead to tighter performance bounds and more efficient (i.e. simpler) proofs. Informally, we may say that Compact Proofs allow us to leverage good MI explanations into tighter and more compact proof bounds. We note that this method allows for finding and evaluating explanations which satisfy many of the Explanatory Virtues: Precise explanations allow for tighter bounds, Accuracy and Simplicity are directly optimised for, and Causal-Mechanistic explanations are generally required for non-vacuous bounds.

## 4.3 DISCUSSION OF EXPLANATORY VALUES

Table 1 shows that some Explanatory Virtues are consistently valued highly across different methods. However, all current interpretability methods could be improved on some dimension to be more likely to produce human-understandable and useful explanations. In particular, we suggest that methods which produce or appeal to nomological principles and which unify accounts of neural network behaviour are likely to be increasingly successful.

---

[9] In general, we might be interested in bounding metrics which are to be minimised (e.g. loss) rather than maximised (e.g. accuracy and reward). In that case we may seek upper bounds rather than lower bounds but the argument is otherwise analogous.

[10] Appendix C provides an example of one basic proof strategy which is computationally expensive but provides a tight bound. This strategy is known as the *brute force proof* (Gross et al., 2024) and corresponds to the *straightforward, Implementation-level explanation* (Ayonrinde & Jaburi, 2025).

## 5   THE ROAD AHEAD

The term Mechanistic Interpretability was coined by Olah et al. (2020) to distinguish itself from previous approaches of neural network interpretability. These previous approaches were not sufficiently grounded in causal abstraction, nor treated the model internals appropriately as representing explanations as intrinsic structure that we would like to uncover (Ayonrinde & Jaburi, 2025; Saphra & Wiegreffe, 2024). The 'Mechanistic turn' in interpretability was a step towards unifying a community around faithful and falsifiable explanations of models. The Explanatory Virtues Framework is a further step in this direction, providing unifying criteria to evaluate explanatory methods. In particular, focusing on the following three virtues would constitute methodological progress for the field:

**1. Simplicity and Compression.**   Swinburne (1997) argues that simplicity is a key virtue of good explanations and can provide evidence to the truth of a theory. However, appropriately characterising an explanatory Simplicity measure is currently an open question for interpretability. Early explorations into understanding compression as a key function of explanation can be found in the Compact Proofs literature (Gross et al., 2024) and Parameter Decomposition (Bushnaq et al., 2025). Coalescing around a concept of Simplicity for interpretability would allow different explanations to be rigorously compared on the (accuracy, simplicity) Pareto curve. Such a definition might also naturally encourage further research into the impact of modularity in both neural networks and their explanations (Clune et al., 2013; Filan et al., 2021; Baldwin & Clark, 1999).

**2. Unification and Co-Explanation.**   Hempel (1966) argues that unification is a core driver of scientific progress. Indeed we may see unification as a drive towards compression of explanations where the set of phenomena to be explained is large (Bassan et al., 2024; Bhattacharjee & von Luxburg, 2024). Currently, most methods in interpretability don't seek to co-explain many phenomena using the same building blocks. The MI community has sought to understand the universality (or otherwise) of representations and algorithms across many models with mixed results (Olah et al., 2020; Olsson et al., 2022; Chughtai et al., 2023). However, we may also be interested in modular compositional explanations where the explanatory units are shared not only across models but also across different tasks and domains within a single model, such as (Merullo et al., 2024a;b; Todd et al., 2024; Yin & Steinhardt, 2025). For example, there is evidence that induction heads are reused for many tasks within models and so induction heads perform a co-explanatory function (Olsson et al., 2022).

**3. Nomological Principles.**   Bacon (1620) writes that any science first starts by observations. Then fields tend towards one of two (non-exclusive) paths that Windelband (1894) refers to as the *nomothetic* and *idiographic* approaches. The nomothetic approach seeks to synthesise these early observations into general explanatory theories with nomological principles that are useful for making predictions. Conversely, the idiographic approach focuses on categorising and describing more sets of observations, without necessarily seeking general laws to explain them. Physics is a prototypical nomothetic science; biology is often considered an idiographic science. Idiographic approaches can tend towards *description* rather than *explanation*. For example, we might wonder if interpretability researchers counting up and categorising all the features in a given model's latent space is much different to a biologist naming and describing all the species of beetle in an ecosystem without learning anything about the evolution of these species or how they interact within the environment.

The use of nomological principles can simplify explanations and help to provide a unifying paradigm for MI. Efforts in Developmental Interpretability (Hoogland et al., 2024), the Physics of Intelligence (Allen-Zhu & Li, 2024), Computational Mechanics (Shai et al., 2024), and the Science of Deep Learning (Lubana et al., 2023; Allen-Zhu & Li, 2023) may also produce useful nomological principles for the MI community to adopt in their explanations.

Mechanistic Interpretability has found Causal Abstractions theory to be a useful foundation. We suggest that a further paradigm for Mechanistic Interpretability should take seriously the virtues of good explanations. The Explanatory Virtues allow us to iteratively build better interpretability methods and generate increasingly good explanations of neural networks. Progress in MI may provide insights into AI systems which are useful for increasing the transparency and safety of systems which are deployed widely and/or in critical applications (Bengio et al., 2025; Rivera et al., 2024; Sharkey et al., 2025). We believe that our Explanatory Virtues Framework can help researchers in designing methods which lead to more reliable and useful explanations of neural systems.

## REPRODUCIBILITY STATEMENT

The comparative evaluation of explanation methods presented in Table 1 can be reproduced by applying the Explanatory Virtues Rubric detailed in Table 2. This rubric provides clear criteria for assessing the extent to which different Mechanistic Interpretability methods embody each explanatory virtue. By following the three-level assessment framework (Highly Virtuous, Weakly Virtuous, Not Virtuous) with their corresponding indicators (✓, ●, ✗), researchers can systematically evaluate explanation methods against the Explanatory Virtues Framework. The rubric's structured approach ensures that assessments are based on consistent criteria rather than subjective preferences, allowing for reproducible comparisons between different explanation methods in Mechanistic Interpretability.

## ETHICS STATEMENT

This work focuses on developing a philosophical framework for evaluating explanations in the context of Mechanistic Interpretability of neural networks. As a theoretical contribution, our framework itself does not directly raise ethical concerns typically associated with empirical AI research, such as data privacy, bias, or direct societal impacts. However, we recognize that advances in Mechanistic Interpretability have significant ethical implications.

Better explanations of AI systems, which our framework aims to encourage, can promote transparency, accountability, and trust in AI systems. We note that improved understanding of neural networks through Mechanistic Interpretability may contribute to AI Safety, AI Ethics, and the responsible deployment of AI systems in critical applications. By providing systematic criteria for evaluating explanations, our work supports the responsible development of AI that is interpretable and human-understandable.

We hope this work contributes to the broader goal of developing AI systems that can be meaningfully understood, monitored, and steered by humans.

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

## A  THE EXPLANATORY VIRTUES

| Explanatory Virtue | Importance | Clustering | (MDL) SAEs | Circuits | Compact Proofs |
|---|---|---|---|---|---|
| *Validity* | | | | | |
| **Causal-Mechanistic** | ! | ✗ | ● | ✓ | ✓ |
| *Bayesian* | | | | | |
| Precision | | ● | ● | ✓ | ✓ |
| Priors | | ● | ● | ✗ | ✗ |
| Descriptiveness | | ● | ● | ✓ | ✓ |
| Co-explanation | | ✗ | ✗ | ✗ | ● |
| Power | | ● | ● | ✓ | ✓ |
| *Bayesian & Kuhnian* | | | | | |
| Accuracy | | ✓ | ✓ | ✓ | ✓ |
| Unification | | ✗ | ✗ | ✗ | ✗ |
| *Kuhnian* | | | | | |
| Consistency | | ● | ✓ | ✓ | ✓ |
| **Simplicity** | ★ | ● | ✓ | ● | ✓ |
| **Fruitfulness** | ★ | ● | ● | ✗ | ● |
| *Deutschian* | | | | | |
| **Falsifiable** | ! | ✓ | ✓ | ✓ | ✓ |
| **Hard-to-vary** | ★ | ● | ● | ✓ | ✓ |
| *Nomological* | | | | | |
| Nomological | | ✗ | ✗ | ✗ | ● |

Table 1: An evaluation of MI explanation methods with respect to our Explanatory Virtues Framework as given in Section 3. The virtues which are indispensable for valid Mechanistic Interpretability explanations are highlighted with a !. The virtues that we consider to be the most important for good explanations are highlighted with a ★. Metrics are grouped by their philosophical foundations: Deutschian, Kuhnian, Bayesian, or Nomological. Blue metrics indicate empirical criteria, while orange metrics represent theoretical criteria. Green checks, orange circles and red crosses indicate that the method well-considers, moderately considers, or poorly considers the virtue, respectively. The explanatory case studies that we have considered generally optimise for accuracy, however they vary in their commitment to the virtues of Simplicity, Unification and Nomologicity. In our descriptions of these methods across Section 4, we provide a more detailed analysis of how we assess the virtues of each method and we provide our full evaluation rubric in Table 2.

These explanatory virtues are not necessarily exhaustive nor completely independent of one another.[11] Additionally, some virtues may be in tension with each other. For example, Accuracy may be traded off against Simplicity in some cases. Here we may aim to be at the optimal point of this trade-off on a Pareto frontier.

We hope the reader may agree that our Explanatory Virtues both (1) are important considerations for the evaluation of explanations and (2) are truth-conducive.[12] Thus, these virtues can be a useful guide for theory choice and, more generally, can aid in the developments of new explanatory methods. Mechanistic Interpretability researchers, we argue, ought to value the Explanatory Virtues.

For an explanation to be a *good* explanation in Mechanistic Interpretability, it must first be a *valid* MI explanation. In Section 2.2 we identified valid MI explanations as those which are Model-Level, Ontic, Causal-Mechanistic, and Falsifiable. Validity requires all of the four validity conditions above

---

[11] We detail an additional possible virtue in Appendix G.

[12] That is to say that all else equal explanations which embody these virtues are more likely to be true. We refer readers to Schindler (2018) for a detailed discussion of the truth-conduciveness of many of the virtues we discuss.

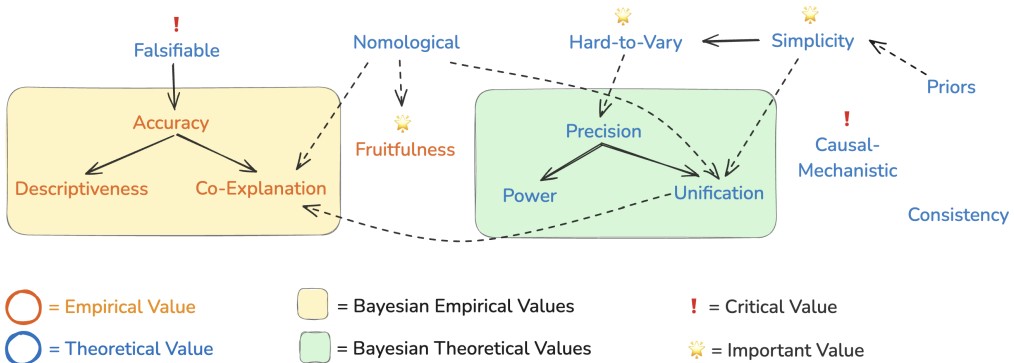

Figure 1: A Directed Acyclic Graph representation of the **Explanatory Virtues Framework** showing the relationships between virtues. Empirical virtues are coloured orange and theoretical virtues are coloured blue. We show the virtues which directly depend on each other with bold arrows ($\rightarrow$) and those which are highly related with dashed arrows ($\dashrightarrow$). The Explanatory Virtues which are essential for any scientific explanation (Falsifiability and Causal-Mechanisticity) to be valid are denoted with an exclamation mark; the most important virtues to decide between explanations (Simplicity, Hard-to-Varyness, and Fruitfulness) are marked with a star. Appendix B details a rubric for assessing explanatory methods. Appendix C provides an example illustrating the importance of Simplicity as an explanatory virtue.

to be met. The Explanatory Virtues, then, allow us to assess the quality of valid MI explanations and provide epistemic reasons to prefer one explanation over another.

## B  THE EXPLANATORY VIRTUES RUBRIC

Table 2:  The rubric for evaluating the Explanatory Virtues of a given explanation (see Figure 1 and section 3). We use this rubric to provide a structured evaluation of explanations as in Table 1.

| Explanatory Virtue | Highly Virtuous | Weakly Virtuous | Not Virtuous |
| --- | --- | --- | --- |
| Icon | ✓ | 🟠 | ✗ |
| Causal-Mechanistic | Generates end-to-end causal explanations | Explains a part of the network and can be used as part of a Causal-Mechanistic Explanation | Generates explanations which are not used for producing end-to-end causal explanations |
| Precision | Rewards explanations that provide precise and risky predictions in a quantifiable way | Partially accounts for precision in explanations, possibly qualitatively | Fails to penalise (or even endorses) overly broad or vague predictions |
| Priors | Explicitly incorporates comparisons with background theoretical priors in the method | Implicitly incorporates background theoretical priors in evaluating explanations | Fails to appropriately incorporate background theoretical priors |

Table 2: The rubric for evaluating the Explanatory Virtues (continued)

| Explanatory Virtue | Highly Virtuous | Weakly Virtuous | Not Virtuous |
|---|---|---|---|
| Descriptiveness | Prefers explanations which clearly analyse detailed, component-wise prediction quality in high fidelity, capturing the essential characteristics of each data point | Only partially tangentially analyses individual data point fit, mostly focusing on overall aggregated fit | No analysis of how the data points fit the explanation in isolation at all |
| Co-Explanation | Assesses the ability of explanations to account for multiple observations together, rewarding measures that emphasise integrated, joint predictive performance. | Has the potential to incorporate some aspects of joint explanation but does not fully reward coherent integration across diverse data points in its currently practised form | Evaluates each data point in isolation, ignoring the value of linking multiple observations. |
| Power | Strongly values approaches that produce highly constraining predictions (especially about observations considered in isolation), penalising methods that allow too many plausible alternatives | Provides moderate emphasis on constraining predictions, allowing for some uncertainty. | Assigns no weight to the predictive force of the explanation |
| Accuracy | Quantitatively rewards explanations that fit the data with minimal error, especially does so with reference to both the precision and recall where relevant | Qualitatively rewards explanations that seem to fit the data well subjectively | Does not distinguish between explanations that fit the data well or poorly leading to evaluations that tolerate significant errors |
| Unification | Measures how well a single evaluation framework can account for diverse observations, emphasizing integrated, unified explanations | Has the potential to recognise some unification even if in a limited or fragmented way or if this is not a typical application of the method | Places no weight on a unified account rather than a disjunction of accounts |
| Consistency | Requires internal coherence within the explanation and multiple instances of running the same explanation method | Mostly internally consistent but probabilistically can provide inconsistent explanations | Places no weight on the internal consistency of generated explanations |
| Simplicity | Evaluates explanations based on a conciseness or K-complexity simplicity metric rewarding simpler explanations | Partially considers a weak form of simplicity such as parsimony | Neglects simplicity as a factor, encouraging highly complex and complicated explanations |
| Fruitfulness | Rewards explanations that predicted new, testable phenomena even with adversarially chosen test data from a close data distribution | Rewards explanations that predict novel phenomena even from the same data distribution | Assesses only current data fit with no train-val-test split at all |

Table 2: The rubric for evaluating the Explanatory Virtues (continued)

| Explanatory Virtue | Highly Virtuous | Weakly Virtuous | Not Virtuous |
|---|---|---|---|
| Falsifiable | Requires that explanations yield clear, testable predictions and penalises those that could not be refuted under counterfactual data. | - | Fails to consider whether explanations can be empirically refuted, rewarding unfalsifiable evaluations. |
| Hard-to-vary | Rigorously assesses the robustness of explanations, rewarding those evaluations where small modifications would lead to significant performance degradation. Checks for interdependencies among components to ensure that each part is essential and load-bearing | Makes limited effort to avoid ad-hoc explanations but doesn't fully address how hard-to-vary the explanations are | Does not account for the ease of altering explanations and consistently produces explanations that are easily tweaked without loss of predictive power |
| Nomological | Explicitly integrates established general laws and principles, favouring evaluations that connect to a broader nomological framework or reusing laws in multiple places across the explanatory theory | Implicitly appeals to some non-generic laws but such a connection may be indirect and not well utilised | Ignores links to universal principles and attempts to focus on explaining the data without any reference to more general theoretical principles |

Explanatory virtues are criteria for theory choice: they help researchers decide which methodological approaches to pursue. We provide a rubric for non-subjectively evaluating whether an explanatory method embodies a virtue in Table 2.

Note that this framework is agnostic to interpretability methods and could be applied to methods from other non-Mechanistic strands of interpretability. However, we focus on Mechanistic Interpretability in particular because non-Mechanistic explanations are, by definition, not concerned with producing Causal-Mechanistic explanations (complete end-to-end accounts of model behavior) which we take to be an important aspect of explanations useful for understanding neural networks (Ayonrinde & Jaburi, 2025; Woodward, 2003; Craver, 2007).

## C STRAIGHTFORWARD EXPLANATIONS

Following (Ayonrinde & Jaburi, 2025), we define the *straightforward explanation* of a neural network as follows. Given a neural network $f : X \to Y$ and $x \in X$ such that $f(x) = y$, the straightforward explanation is given by the computational trace of the network on the input x.[13] We note that for any neural network $f$ and sub-distribution $D \subseteq \mathcal{D}$, there exists a straightforward explanation of $f$ on $D$. However, this straightforward explanation is typically not good a explanation in the sense of Section 3 as such explanations are not very concise or illuminating. We would instead like explanations of neural networks that are in terms of the features (or concepts) that the network learned during training and explanations which are compact and useful.

Given Section 3 and Appendix B we may evaluate the straightforward explanation of a neural network using the Explanatory Virtues Framework.

---

[13] In fact, this explanation is a formal proof of the equality $f(x) = y$.

- **Causal-Mechanistic:** The straightforward explanation is Causal-Mechanistic. It decomposes the model into a computational graph, given by the neural network architecture.

- **Precision, Descriptiveness, Accuracy, Power & Falsifiable:** The straightforward explanation fulfills all these criteria, since it is a complete representation of the model.

- **Co-Explanation & Unification:** The straightforward explanation does not fulfill these criteria, since it treats all inputs independently.

- **Priors:** The straightforward explanation does not refer to priors in its interpretation.

- **Consistency:** The straightforward explanation is consistent.

- **Simplicity:** The straightforward explanation is highly complex. There is no compression from the original weights in the explanation given.

- **Fruitfulness:** The straightforward explanation is not fruitful, in that it doesn't provide novel predictions.

- **Hard-to-vary:** The straightforward explanation is not hard-to-vary; modifying single parts of the model (e.g. individual weights) by some small amount will typically not vary the model performance.

- **Nomological:** The straightforward explanation is not nomological as it doesn't provide general laws or principles.

We note that the straightforward explanation is a valid explanation of a neural network: It is Model-level, Ontic, Causal-Mechanistic, and Falsifiable. Further, the straightforward explanation embodies many of the explanatory values. However, we hope the reader will agree that the straightforward explanation is not a *good explanation*. Since, as noted in Section 5, not all of the explanatory values are equally as important, an explanation may embody some of the virtues and yet not be a good explanation.

Researchers who are interpreting a neural network may have different use cases for which they would like an explanation of the model behaviour. To account for these different goals, researchers can make trade-offs between which Explanatory Virtues they value most highly.[14] Overall, however, for an explanation to be a good explanation, we suggest that *Simplicity* and *Fruitfulness* and *Hard-to-Varyness* are the most important values, without which it is difficult to have a good explanation. In this case, the straightforward explanation fails on the virtue of Simplicity.

## D  EXPLANATIONS IN THE WILD, VISUALLY

This section is a visual companion to Section 4. We present a series of figures to elucidate what we mean by each form of explanation and how we choose between two explanations given this method (i.e. Theory Choice (Schindler, 2018)).

---

[14] Choosing the right explanation is a value-laden task (Ayonrinde & Jaburi, 2025).

## D.1  CLUSTERING (ACTIVATIONS OR INPUTS)

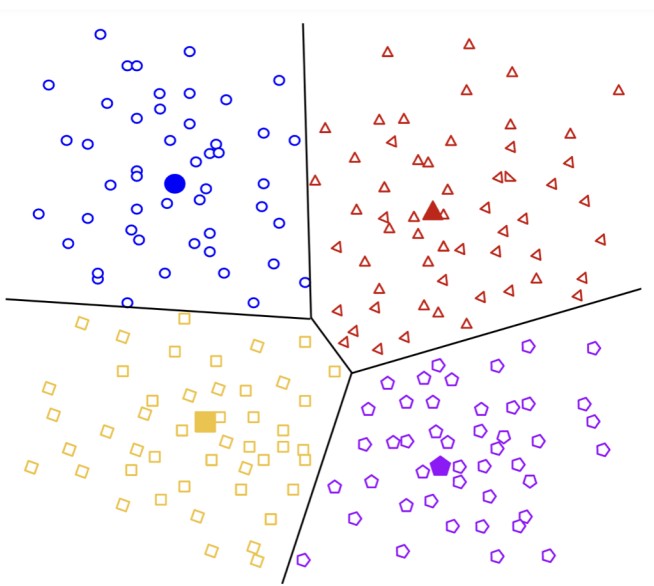

Figure 2: Given some (possibly intermediate) embeddings ($\mathbf{x}$), a clustering explanation can be produced by assigning $\mathbf{x}$ to a cluster $C_i$, where the n clusters partition the input space into disjoint regions. Here $C_1 \cup C_2 \cup \ldots \cup C_n = \mathbb{R}^N$ and $C_i \cap C_j = \emptyset \ \forall i \neq j$. The explanation is then given by taking the behaviour of the model on some cluster representative, or centroid, $\mu_i \in C_i$. We can intuitively see this as performing a quotient operation on the input space, where the model behaviour is approximated by a piecewise constant function. [Image from Google Developers (2025)].

## D.2 Sparse Autoencoder Explanations of Representations/Activations

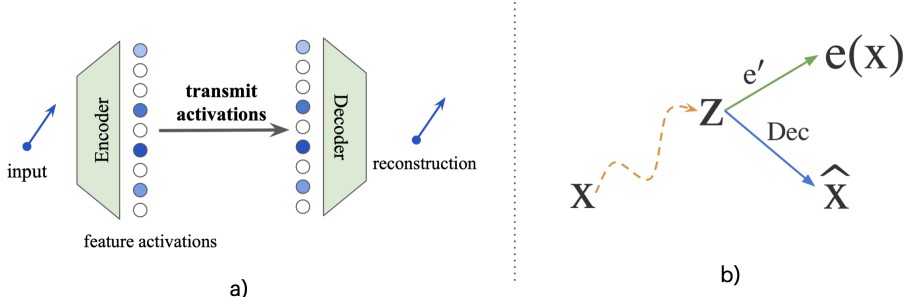

Figure 3: (a) The SAE architecture. An encoder provides some set of latents (or feature activations) in the feature basis. We have some decoder map, Dec, which is a linear combination of the columns of the feature dictionary weighted by the sparse latents. We say informally that these latents *correspond* to the input activations if, under the decoder map, Dec. (b) If $\mathbf{x}$ and $\mathbf{z}$ correspond in the above sense then the natural language explanation of the input activations $\mathbf{x}$ is given as $e(\mathbf{x}) = e'(\mathbf{z})$; that is the explanation of the latents using the automated interpretability process $e'(\mathbf{z})$ (Paulo et al., 2024; Karvonen et al., 2024; Bills et al., 2023; Ayonrinde, 2024). We can measure the mathematical description length (*Conciseness*) of the explanation $e(\mathbf{x})$ as the number of bits required to describe the latents $\mathbf{z}$ (Ayonrinde et al., 2024). [Images from Ayonrinde et al. (2024); Ayonrinde (2024)]

## D.3 Causal Abstraction Explanations of Circuits

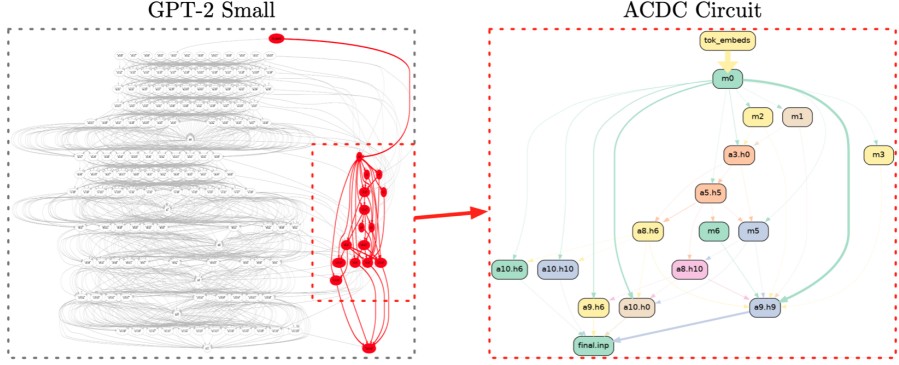

Figure 4: A circuit explanation is a Causal-Mechanistic explanation such that the circuit C is a constructive abstraction of a neural network's computational graph M if there exists a partition the variables in M such that each high-level variables in C correspond to a low-level partition cell in M and interventions on M correspond to interventions on C. For example in Figure 4 *Left* (Conmy et al., 2023), the IOI circuit (Wang et al., 2023) (highlighted in red) is recovered from the computational graph of GPT-2 Small. [Image from (Conmy et al., 2023)].

## D.4 COMPACT PROOFS

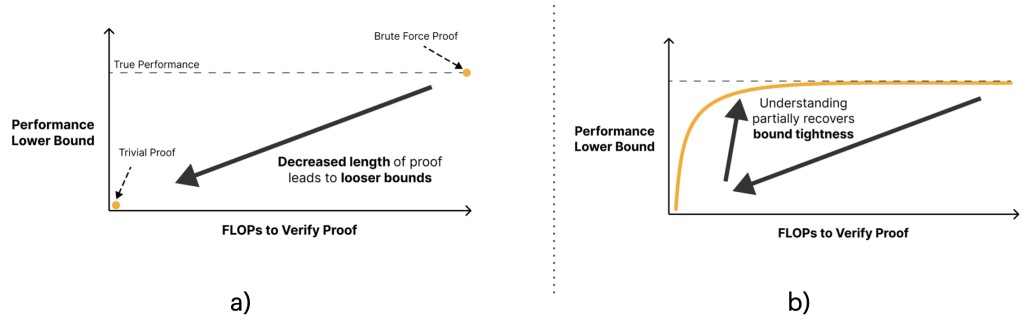

a)  b)

Figure 5: (a) Compact Proofs evaluate explanations on two metrics, their compactness (FLOPs to Verify Proof) and their accuracy (Model Performance Lower Bound). These two metrics can be assessed on a Pareto frontier. (b) A good explanation should push the frontier towards the upper left corner (i.e. more accurate and compact proofs). [Image from Gross et al. (2024).]

## E EXAMPLES OF EXPLANATIONS

In this section, we provide some intuitive examples and non-examples of Explanations which satisfy the criteria that we outline above. The case studies in Section 4 are examples within Mechanistic Interpretability and Machine Learning; our examples here are non-technical illustrations.

### E.1 EXAMPLES OF EXPLANATION TYPES

#### E.1.1 ONTIC EXPLANATIONS

*Question*: Why did the pen fall off the desk?

**Causal-Mechanistic But Not Ontic Explanation.**

> The pen fell off the desk because the aether pushed the bottle and then the bottle pushed the pen off the desk.

This explanation is Causal-Mechanistic in the sense that one thing happens after another and causes the next. However, if we do not believe that the aether is a real entity then this explanation cannot be considered an Ontic Explanation.

—

*Question*: Why is the cube heavy?

**Ontic But Not Causal-Mechanistic Explanation.**

> The cube is heavy because it is made up of tungsten atoms.

This explanation is Ontic as the entities involved in the explanation are real entities. However, it is not Causal-Mechanistic as there is no step-by-step explanation without gaps.

#### E.1.2 STATISTICALLY-RELEVANT EXPLANATIONS

Consider the explanation:

> Ice cream sales are higher on days when there are more shark attacks. If there's a shark attack reported, we can predict with 85% confidence that ice cream sales will be above average that day.

This explanation is purely in terms of statistical correlation rather than causation. There is no explication of any underlying causal mechanism, which might involve both phenomena being causally downstream of hot weather and/or more beach visitors. We could perform interventions to test this hypothesis.

### E.1.3   NOMOLOGICAL EXPLANATIONS

*Question*: Why does a metal rod expand when heated?

**Nomological but not Causal-Mechanistic explanation.**

> The rod expands because it follows the natural law that all metals expand when heated, as described by the coefficient of thermal expansion.

This explanation references a general law of nature without getting into the underlying mechanism.

**Causal-Mechanistic Explanation.**

> The rod expands because its metal atoms vibrate more vigorously when heated, which increases their average spacing. This increased spacing leads to an overall increase in the rod's length.

This details the physical mechanism causing the expansion.

### E.2   EXAMPLES OF EXPLANATORY VALUES

### E.2.1   PRECISION, POWER AND UNIFICATION

Consider one explanation of what happens to objects when they are dropped:

> When an object is dropped, it falls to the ground due to the force of gravity.

compared to the more **precise** explanation:

> Objects fall toward Earth at a rate of 9.8 meters per second squared, with slight variations depending on altitude and latitude.

The latter explanation rules out more possibilities than the former. When we see that an object is dropped, armed with the second explanation, we are able to rule out the possibility that the object will fall at a different rate as well as the possibility that it will rise into the air.

Precise explanations make *narrow* and *risky* predictions.

**Unification.**   An explanation is **unifying** if it purports to explain multiple disparate observations. The Central Dogma in molecular biology states that genetic information flows only in one direction, from DNA, to RNA, to protein, or RNA directly to protein (Crick, 1970). This theory operates as a unifying explanation which narrows the space of possibilities for a wide range of biological phenomena.

### E.2.2   CONSISTENCY

Consistent explanations contain no internal contradictions.

*Question*: Why did Alice miss the important meeting this morning?

**Inconsistent Explanation.**   Alice, being a forgetful person, forgot that the meeting was happening and simultaneously Alice deliberately skipped the meeting to avoid a confrontation.

**Consistent Explanation.** Alice was out of the office for a vacation and missed the meeting.

As we increase the unification/scope of explanations, we sometimes introduce inconsistencies. For example, as we look to unify Quantum Mechanics and General Relativity, two explanations which are internally consistent on their own, we find that they are inconsistent with each other.

### E.2.3 SIMPLICITY

Occam's Razor states that when faced with competing explanations, one should select the explanation that is the simplest. This heuristic was first formulated in terms of parsimony, but we might also extend the sense of simplicity here to conciseness (Shannon complexity) or K-complexity (Kolmogorov complexity) as more appropriate measures of simplicity.

The Ptolemaic explanation:

> The Earth is at the center of the universe, with the planets, the sun, and stars orbiting around Earth. There are many epicycles which explain the retrograde motion of the planets (planets moving backwards in the sky).

is more complex than the Copernican explanation:

> The sun is at the center of the solar system and the planets orbit the sun in ellipses.

Even though both explanations could fit the data, we ought to prefer the Copernican model according to Occam's Razor and our Explanatory Virtue of Simplicity.

Wojtowicz & DeDeo (2020) give a sobering example of the dangers of not sufficiently valuing simplicity in explanation in their analysis of conspiracy theories. Such theories are often "abnormally co-explanatory and descriptive ..., account for anomalous facts which are unlikely under the 'official' explanation ..., show how seemingly arbitrary facts of ordinary life are correlated by hidden events ..., and describe a unified universe where everything is correlated by a network of hidden common causes." A primary reason that such conspiracy theories are not typically good explanations is that they are not *simple*: there's often a large amount of complexity and ad-hoc reasoning to explain contradictory evidence and the reason for why the cover-up has yet to come to light.

### E.3 FALSIFIABILITY AND HARD-TO-VARYNESS

Popper (1935) argues against the pseudoscientific theories of Marx, Freud, and Adler on the grounds that they are not falsifiable. That is to say, there exists no observation that could be made that would contradict the theory and cause its proponents to abandon it. For a theory to be falsifiable it must make some concrete predictions about the world that could in principle be tested.

Consider the following three explanations for why there are seasons (adapted from Deutsch (2011)):

**Not Falsifiable.**

> The seasons change when Zeus feels like it.

This explanation is not falsifiable because it does not make any predictions. If there were no seasons one year, then it would not be a mark against the theory.

**Falsifiable but Not Hard-To-Vary.**

> Demeter (the Greek Goddess) negotiates a deal with Hades such that her daughter Persephone visits Hades once a year. When Persephone is with Hades and not with her mother, Demeter is sad and the world becomes cold.

This explanation does make a concrete prediction: the seasons will change exactly once a year. Another prediction that follows is that winter (the period of cold where Persephone is with Hades) should happen everywhere on Earth at the same time. This explanation is falsified by the fact that the seasons are at different times in Australia to in Athens. The explanation is not very Hard-to-Vary however. We could easily change any of the characters or mechanisms involved in the theory and keep the same predictions.

**Falsifiable and Hard-To-Vary.**

> The Earth's axis of rotation is tilted relative to the plane of its orbit around the
> sun. Hence for half of each year the northern hemisphere is tilted towards the sun
> while the southern hemisphere is tilted away, and for the other half it is the other
> way around. Whenever the sun's rays are falling vertically in one hemisphere (thus
> providing more heat per unit area of the surface) they are falling obliquely in the
> other (thus providing less heat).

This explanation is both falsifiable and hard-to-vary. All of the details of the theory play a functional
role and cannot be easily changed. The axis-tilt theory also (correctly) predicts the fact that the
seasons are reversed in the northern and southern hemispheres.

### E.4 (MUNDANE) ACCURACY AND FRUITFULNESS (NOVEL SUCCESS)

Explanations have Mundane Accuracy insofar as they correctly account for the phenomena they aim
to explain. Conversely explanations are Fruitful if they predict new phenomena that were not available
to the explainer at the time of coming up with the explanation. Being able to predict and explain new,
previously unobserved phenomena that are later confirmed (as in Fruitfulness) is typically considered
more valuable than merely explaining known phenomena (as in Mundane Accuracy).

Einstein's General Relativity predicted that light would bend around massive objects like the sun
(Einstein, 1916). In 1919, during a solar eclipse, Arthur Eddington observed that starlight passing
near the sun was indeed deflected by precisely the amount Einstein had predicted (Dyson et al.,
1920; Kennefick, 2021). Given that the phenomenon of light bending around massive objects was
previously unknown, this was a novel empirical success for Einstein's theory. This can increase our
credence in Einstein's theory because the prediction was made before the observation, was precise and
quantitative in an unknown domain and the observations matched the prediction with high accuracy.

### E.5 CO-EXPLANATION AND DESCRIPTIVENESS

Explanations can be purely *descriptive*, in which case they account well for the phenomena they aim to
explain but do not connect with other explanations. Alternatively, explanations can be *co-explanatory*,
unifying phenomena that were previously thought to be distinct.

**Descriptive but Not Co-Explanatory.**

> Electricity involves the movement of charges and produces effects such as static
> attraction, lightning, and electrical current. Magnetism, on the other hand, involves
> the attraction or repulsion between certain materials like lodestone and iron, and
> manifests in the behavior of compasses pointing north.

**Co-Explanatory.**

> Electricity and magnetism are manifestations of a single underlying electromagnetic
> force. A changing electric field produces a magnetic field, and a changing magnetic
> field produces an electric field. Moving electric charges create magnetic fields,
> while moving magnets induce electric currents.

## F A COHERENCE FORMULATION OF ADHOCNESS

(Schindler, 2018) also gives an adhocness test for explanations which can identify those which are
the result of a post-hoc epicycle added to an easy-to-vary explanation. For Schindler, an explanation
is adhoc if the modification $\Delta$ which it corresponds to is some additional hypothesis H (which we
can think of as being added in order to accommodate some awkward-to-explain data $\mathbf{x}_I$) and two
conditions are met:

1. H explains $\mathbf{x}_I$. That is $\mathbb{P}(\mathbf{x}_I|E, H) > \mathbb{P}(\mathbf{x}_I|E)$.

2. Neither the original explanation E nor background theories B give evidence for H. That is $\mathbb{P}(H|E,B) < \mathbb{P}(H)$.

We define an adhocness metric as Adhoc $= \mathbb{P}(H) - \mathbb{P}(H|E,B)$ where larger ad-hocness values are more adhoc and dispreferred.

## G  LOCAL DECODABILITY AS AN EXPLANATORY VIRTUE

Another virtue that we may consider for highly unifying explanations is local-decodability. Locally decodable explanations allow for retrieval and use of some small segment of the explanation without querying the whole explanation, analogously to locally-decodable error-correcting codes (Yekhanin et al., 2012). This is important as we would like not only for our explanations to have information compression (concise representation) but also information accessibility (the ability to retrieve specific subparts quickly). In practice, an explanation of network which is compressed but not locally-decodable requires significant computational resources to query and is not useful for human understanding.[15] The Independent Additivity condition from Ayonrinde et al. (2024) is an example of a local-decodability condition in Mechanistic Interpretability. V-Information (Xu et al., 2020) provides a useful analogy for local-decodability in Machine Learning.

## H  COMPARISON TO MECHANISTIC INTERPRETABILITY BENCHMARK

Mueller et al. (2025) recently proposed Mechanistic Interpretability Benchmark (MIB), which is intended to test whether interpretability methods achieve improvements over simple baselines. Their benchmark focuses on two tracks:

1. **Circuit Localisation**: comparing methods that find subnetworks within a model which are most important for performing a task (e.g., attribution patching or information flow routes) and

2. **Causal Variable Localisation**: comparing methods that produce vectors which correspond to a model feature and are causally relevant for a given task.

To align with the framework in Chalmers (2025), we may think of Circuit Localisation as aiming to test for *Algorithmic (Mechanistic)Interpretability* and Causal Variable Localisation as aiming to test for *Conceptual Interpretability*. In our terminology, Causal Variable Localisation does not provide explanations which are Causal-Mechanistic in nature (as they do not produce end-to-end explanations of model behaviour) but they provide useful building blocks for Causal-Mechanistic explanations.

We believe that MIB is a valuable step forward for the MI community because for methods that have the same inputs and affordances, they can be directly compared using their benchmark with respect to the downstream tasks that the authors list. To the extent that these tasks are indeed representative of the goals that we have for MI methods then their comparison is highly useful.

However, there are some downsides to the approach that Mueller et al. (2025) take. In particular: Some of the methods that the authors compare are not directly comparable as e.g. they compare supervised and unsupervised methods. The explanations are not all complete end-to-end explanations of the model's internal algorithms and so many do not focus on algorithmic interpretability, which we believe to be the core of Mechanistic Interpretability (Ayonrinde & Jaburi, 2025; Olah et al., 2020). MIB assumes a particular form of Simplicity, Parsimony, which is known to have problems as detailed in Section 3.2. This severely hampers their ability to correctly evaluate how simple an explanation is. Similarly, Mueller et al. (2025) do not take into account the benefits of having explanations which unify observed phenomena or utilise nomological principles. We believe that this may implicitly encourage researchers to produce explanations which do not reuse components and hence are ultimately less human-understandable and less able to stand on the shoulders of previous useful explanations.

---

[15] Local decodability is measured in query complexity: the number of queries required to recover 1 bit of the message (explanation). Conciseness and query complexity are known to be inversely proportional but the exact fundamental limit on their relationship is currently unknown.

Our approach differs because we ask the core question: "if I'm creating a new method for creating explanations for interpretability, which properties should my method value?" This framing has the advantage of picking out the properties for which if a method selects for explanations that perform well with respect to those properties, the explanation is likely to be a faithful and useful explanation for researchers. Note that our criteria are not intended to say that Method A is uniformly (e.g.) accurate as Method A may be more or less accurate on different models/tasks. We are instead asking the question of whether Method A is set up to value Accuracy and would, on the margin, prefer more explanations which are more accurate. In this way we are evaluating explanatory methods rather than the output of an explanatory method on a specific task.

We believe that our framework is a useful complement to the MIB paper which goes beyond evaluating on a relatively narrow set of tasks and gives researchers practically useful criteria to check that their methods for choosing between evaluations captures. In the MI stack, we might see the Explanatory Virtues Framework (EVF) as sitting in a complementary position to MIB in that we may use the EVF to diagnose the MIB and understand where it may not effectively distinguish between explanations. Where the EVF evaluates whether explanation-generating methods have the right design principles to produce good explanations, MIB evaluates the outputs of those methods on specific tasks. Our framework operates at the meta-level — we ask "does this method tend to produce explanations with desirable properties?" rather than "how well does this specific explanation perform on task X?"

Our framework has three core points of complementarity with MIB: Firstly, it can help diagnose why certain methods succeed or fail in MIB's benchmarks. Secondly, our framework can help researchers design better methods that would then perform well on benchmarks like MIB. Thirdly, our framework allows researchers to see the drawbacks of MIB and where good performance on MIB and good explanations of neural networks may come apart. This helps avoid the Goodharting of MIB at the expense of good explanations. Here we can also use our framework to design better versions of MIB in the future which are better aligned with our true goals as interpretability researchers.

## I    THE IDENTIFIABILITY OF MECHANISTIC INTERPRETABILITY

Recently, Méloux et al. (2025) showed that different networks exhibiting the same behaviour can have different underlying implementations on the computational substrate. This is analogous to multiple realisability in the Philosophy literature (Bickle, 2020). We find their work to be a particularly striking and clear example of this multiple realisability phenomena applied to MI.

We note the complementarity with our framework. We are stating that for any two possible explanations of implementations in a single model both analysing the same phenomena, we would like to be able to pick out better rather than worse explanations (which can be empirically achieved by seeking explanations which are virtuous in the sense given in Section 3).

Méloux et al. (2025) highlight the fundamental importance of Mechanistic Interpretability focusing on explanatory faithfulness rather than merely behavioural faithfulness. Without explanatory faithfulness, we would not be able to express or understand the distinction between different circuit algorithms which compute the same result. As a classical computing example, we can think of this as being able to distinguish between different sorting algorithms, such as quicksort and mergesort, which both produce the same sorted output but do so via different computational processes.

## J    COMPACT PROOFS GLOSSARY

This section provides a glossary for terms in Section 4.2. We refer readers to (Gross et al., 2024) for a more detailed discussion of the Compact Proofs Evaluation Methodology.

- **Proofs:** are a sequence of statements in a formal language which are taken as a logically valid argument for why the statement to be proved must be true. For example, $\forall x \in \mathbb{N} : x + 1 = 1 + x$ is a formal statement which can be formally verified using a formal proof system such as Coq (The Coq Development Team, 2023) or Lean.
- **Compactness:** The length of the proof that captures the cost of running the computations it postulates. We can quantify the length of a proof using two metrics:
    1. The precise number of FLOPs required to verify the proof.

2. Its asymptotic complexity in terms of specific input parameters.

For example, verifying that $x_1 + x_2 + ... + x_n = x_{sum}$ for fixed $x_1, x_2, ..., x_n, x_{sum}$ numbers in (some finite precision format) requires $n - 1$ FLOPs to verify the proof and scales asymptotically with $\mathcal{O}(n)$. A proof with shorter length is said to be more compact.

- **Bounds of model performance:** Performance on a model is a quantifiable metric $f : W \to \mathbb{R}$ from the weight space $W$ of the model to the real numbers. This can refer to e.g. the model's accuracy on a data distribution $\mathbb{D}$ (such as the test or training set). A bound $b : W \to \mathbb{R}$ is a function which lower bounds the model performance, such that for all $w \in W, b(w) \leq f(w)$. [16] For example, we may want to prove that models subject to a specific mechanistic property (e.g. an induction head) will achieve at least a certain accuracy on a test set (e.g. all sequences of the form $...AB...A[B]$).

In practice, proofs for bounds of model performance with weights $w \in W$ consist of two parts:

1. (General theorem) A proof $Q_1$ of a theorem of the form "$w \in W, b(w) \leq f(w)$".

2. (Specific computation) A computational trace $Q_2$ which computes $b(w)$ for a specific $w \in W$.

This gives us the guarantee we need: For our concrete weights $w_0 \in W$, $Q_1$ guarantees that the number we will compute $b(w_0)$ (through some algorithm) is indeed a lower bound of $f(w_0)$. Then $Q_2$ guarantees that the we ran the algorithm correctly to compute $b(w_0)$.

The length of the proof is the sum of the lengths of $Q_1$ and $Q_2$. We expect the length of $Q_2$ to dominate as we need to perform many computations with high-dimensional tensors.

## K  THE FCM CRITERIA FOR CIRCUITS

For $C$ a proposed circuit and $M$ the model, the **Completeness** criterion states that for every subset $K \subset C$, the incompleteness score $|F(C \setminus K) - F(M \setminus K)|$ should be small. Intuitively, a circuit is complete if the function of the circuit and the model remain similar under ablations. Conversely, the **Minimality** criterion states that for every node $v \in C$ there exists a subset $K \subseteq C \setminus \{v\}$ that has high minimality score $|F(C \setminus (K \cup \{v\})) - F(C \setminus K)|$. Intuitively, a circuit is minimal if it doesn't contain components which are unnecessary for the function of the circuit.

Note that, corresponding to our Explanatory Virtues, the (behavioural) Faithfulness of an explanation is an Accuracy property. Completeness looks to provide additional evidence towards Accuracy towards explanatory faithfulness (Ayonrinde & Jaburi, 2025). Minimality is a Simplicity property.

## L  APPLYING THE EXPLANATORY VIRTUES FRAMEWORK

In practise we hope that our Explanatory Virtues Framework can be used by MI researchers when designing new interpretability methods and evaluation metrics. Existing examples of the value of the framework include the MDL-SAE method from Ayonrinde et al. (2024) and Wu et al. (2024)'s unification of explanations for Group Operations.

The insight of the MDL-SAE paradigm was that in changing from Parsimony to Shannon complexity as the measure of Simplicity for SAEs, many of the existing problems with SAEs were alleviated (see Section 4.1.2). We encourage researchers to focus on the Simplicity metric that is best aligned for their task and note that Parsimony (while implicitly the most popular measure of Simplicity in the MI literature) is a poor guide to Simplicity. Parsimony treats intuitively highly complex objects and very simple objects both equivalently as "entities" and simply counts them up without nuance. (Baker, 2022) provides a discussion of the downsides of Parsimony as a measure of simplicity.

Wu et al. (2024) demonstrated our framework's utility by applying the Compact Proofs methodology to three competing interpretations. They found that two interpretations failed to produce non-vacuous bounds (indicating poor Accuracy and Simplicity), while their interpretation succeeded. This exemplifies how our framework can resolve explanatory conflict.

---

[16]Depending on the metric, we may also consider upper bounds, where $b(w) \geq f(w)$ for all $w \in W$.

