# OpenReview forum: "Evaluating Explanatory Evaluations: An Explanatory Virtues Framework for Mechanistic Interpretability"
_ICLR.cc/2026/Conference — ICLR 2026 Conference Withdrawn Submission_

### Official Review · Reviewer_JRo2 · 2025-10-31

**Soundness:** 3
**Presentation:** 3
**Contribution:** 2
**Rating:** 4
**Confidence:** 3

**Summary:**

This paper introduces the Explanatory Virtues Framework (EVF), a pluralist philosophical approach for evaluating Mechanistic Interpretability (MI) methods in neural networks. Drawing on Bayesian, Kuhnian, Deutschian, and Nomological perspectives from the philosophy of science, the authors formalize several “explanatory virtues” (\simplicity, unification, falsifiability, nomologicity) and argue that current MI methods only partially capture them. They analyze methods such as clustering, sparse autoencoders, causal circuits, and Compact Proofs (Wu et al., 2024), concluding that Compact Proofs best align with these virtues. The work primarily contributes a conceptual and philosophical synthesis rather than new technical results.

**Strengths:**

- Provides a philosophically grounded synthesis of evaluation principles for MI and XAI, connecting interpretability to broader scientific reasoning.
- Offers a unifying vocabulary for comparing disparate MI evaluation methods, which can help clarify current methodological gaps.
- The case studies (SAEs, circuits, Compact Proofs) make the discussion concrete and show how philosophical virtues translate into MI practice.
- Well written, structured, and formatted, with a coherent narrative linking philosophy of science to AI safety and explainability.

**Weaknesses:**

- The work lacks substantive novelty or technical contribution, as it mainly reviews and organizes prior ideas rather than introducing new methods or empirical results (see lines 70–72, where Compact Proofs are described as already embodying many of the Explanatory Virtues).
- The analysis of Compact Proofs (Wu et al., 2024) evaluates an existing framework rather than extending or formalizing it further, raising questions about incremental contribution.
- No quantitative or experimental validation of the framework is provided; claims of systematic evaluation remain conceptual.
- Much of the content reads as a philosophical position or survey paper rather than a technical ICLR submission, with limited algorithmic or empirical substance.

Suggestion: Figure 1 is interesting (particularly for the visual reader), though I'm not sure why it's referred to as a DAG in the caption. In any case, if this Figure were somehow merged with (condensed versions of) the technical definitions presented in Sections 3.1-3.4, and brought into the main body, that may improve readability throughout.

**Questions:**

- Can the authors illustrate how these virtues could be implemented quantitatively, e.g., through measurable proxies or scoring metrics?
- Since Compact Proofs already embody many of these virtues, what new evaluative insight does EVF provide beyond existing analyses?
- Will the proposed rubric or framework be released in a form usable by the MI community for reproducible evaluations? Perhaps something like a datasheet for dataset or model cards?
- How broadly do the authors envision the framework applying beyond MI—for instance, to model-agnostic or post-hoc explainability methods?

---

### Official Review · Reviewer_fRBQ · 2025-11-01

**Soundness:** 3
**Presentation:** 3
**Contribution:** 3
**Rating:** 4
**Confidence:** 4

**Summary:**

Many explanatory techniques have recently emerged in the field of Mechanistic Interpretability (MI), but while preparatory work has been achieved in that regard, no consensus yet exists as to what constitutes a "good explanation" or a "good explanation-generation method". In addition, some explanatory methods may generate different explanations for a single model and task, with no clear way to choose between them. To solve this, the authors introduce the normative Explanatory Virtues framework, a set of desirable properties of explanations (or explanation methods) taken from four different approaches in the philosophy of science (Bayesian, Kuhnian, Deutschian and Nomological). They provide a rubric that defines satisfaction criteria for these virtues, and compare several explanatory methods (clustering, SAEs, circuits and Compact Proofs) under the lens of this rubric. Out of the evaluated methods, the authors find that Compact Proofs satisfy the most virtues. The listed virtues are also intended to help MI researchers develop better explanation methods and evaluation metrics.

**Strengths:**

- The identified problem is important and a common point of contention in modern MI, as identified by the authors. The amount of available techniques in MI has exploded, but progress is slowed by the lack of consensus on what it means for an explanation to be good, or how to compare competing explanations. This work provides a framework to assess these questions and ground future MI methods into well-accepted virtues from the philosophy of science.
- The paper is well-written, structured, and argued. The authors do a good job of introduce complex philosophical concepts to an ML audience. The framework is systematically developed, virtues well-defined and grounding in the philosophy of science. Some virtues are defined using formalizations (the Bayesian ones). The detailed rubric included in Table 2 and the illustrative examples in the Appendix make the framework very clear.
- The application of the framework to existing MI methods in Section 4 is rather compelling and shows the immediate applicability of the framework. For instance, the authors identify the importance in shifting from Parsimony to Correctness in the case of SAEs, and identify areas of improvement needed for circuit research (unification and nomologicity).

**Weaknesses:**

- The work is crucially missing a discussion on prior work on desirable explanation properties in MI. Comprehensive frameworks like the "Co-12 properties" proposed by Nauta et al (2022) provide a well-established vocabulary for discussing the quality of explanations, with many of their properties having direct counterparts in the Explanatory Virtues framework. Other (albeit less detailed) such frameworks can be found field-specific reviews ("Essential properties and explanation effectiveness of explainable artificial intelligence in healthcare: A systematic review", "How Should AI Decisions Be Explained? Requirements for Explanations from the Perspective of European Law", "Finding the right XAI method — A Guide for the Evaluation and Ranking of Explainable AI Methods in Climate Science") as well as applied works such as "What Makes a Good Explanation?: A Harmonized View of Properties of Explanations". Regrettably, the authors do not define their contribution in relation to these existing frameworks. As a result, researchers familiar with the XAI literature might see the paper as reinventing established concepts rather than building upon them. This is by far the most important flaw of the paper, which makes me lean towards rejection.
- One weakness of this work is the difficulty in operationalizing and quantifying some of the proposed virtues, which limits its immediate practical usefulness. While the Bayesian virtues are formalized, some concepts (e.g. Hard-to-Varyness, Unification, Nomologicity) are mostly qualitative. The rubric in Table 2 is useful, but it relies on categorical judgments (Highly/Weakly/Not Virtuous). A discussion of the challenges of developing more quantitative measures for these virtues would help strengthen this paper. Another option is providing more concrete guidance on how to apply the rubric to avoid ambiguity.
- The paper is entirely theoretical. This is appropriate for a conceptual contribution of this nature, but it likely limits its visibility to the targeted MI community. The impact of the paper could be vastly amplified by even a small, demonstrative use case. For example, the authors could show how to apply Explanatory Virtues to modify an existing MI method like SAEs or circuit discovery in a non-trivial way, and then empirically demonstrate that this leads to explanations that are measurably better (more robust, generalizable, useful for a downstream task like model editing...).

**Questions:**

- The authors convincingly argue that "Nomological Principles" are neglected in practice. What could a "general law" or "universal principle" for neural networks look like? Are there any early candidates from the literature?
- The formalization of "Hard-to-Varyness" is elegant, but its utility depends on the choice of the complexity measure. What are the authors' thoughts on appropriate choices for that measure in the context of MI?
- How do the authors see their framework interacting with newer benchmarks like the MI benchmark mentioned in the appendix? The authors position the EVF at a "meta-level," but the framework is only as useful as its impact on future research. Could the virtues be used to design better, more comprehensive benchmark tasks that go beyond simple localization?

---

### Official Review · Reviewer_uR5e · 2025-11-04

**Soundness:** 2
**Presentation:** 2
**Contribution:** 1
**Rating:** 2
**Confidence:** 4

**Summary:**

This paper proposes the Explanatory Virtues Framework (EVF) as a philosophical and theoretical foundation for evaluating explanations in Mechanistic Interpretability (MI). Drawing from the Philosophy of Science, the authors aim to define “what makes a good explanation.” They formalize a set of Explanatory Virtues (such as accuracy, simplicity, unification, falsifiability, and nomologicity) and use them to analyze common MI methods (e.g., clustering, sparse autoencoders, circuit analysis, and compact proofs). The paper argues that compact proofs best embody these virtues and suggests future research directions emphasizing simplicity, unification, and nomological principles.

**Strengths:**

1.	The paper provides a taxonomy of explanatory virtues, offering an evaluation of interpretability methods.
2.	The review of clustering, SAEs, causal circuits, and compact proofs serves as a concise overview of current MI approaches.

**Weaknesses:**

1.	The paper offers no new formulation, no implemented algorithm, and no quantitative evaluation, making it philosophical rather than scientific in nature.
2.	The claims about the framework’s effectiveness (e.g., compact proofs being superior) are purely conceptual, with no experiments, case studies, or metrics applied to real interpretability outputs.
3.	Many of the proposed “virtues” (e.g., accuracy, simplicity, falsifiability) are well-known scientific virtues. The integration of these ideas into MI lacks formal innovation or clear methodological advancement.
4.	The work does not engage deeply with quantitative interpretability metrics (e.g., faithfulness, sparsity, stability) or explain how EVF would modify or replace these measures in practice.
5.	There is no demonstration that EVF can guide researchers toward better MI models or yield measurable interpretability improvements.

**Questions:**

1.	How can the proposed Explanatory Virtues Framework be operationalized for actual interpretability evaluation? Can it produce measurable scores or comparative results?
2.	What specific methodological advances (algorithms, datasets, or metrics) would EVF inspire for future research?

---

### Official Review · Reviewer_r4W2 · 2025-11-07

**Soundness:** 1
**Presentation:** 1
**Contribution:** 1
**Rating:** 2
**Confidence:** 5

**Summary:**

The paper aims to offer a meta-theory for evaluating different theories of explanations that occur within the Mechanistic Interpretability approach to explainable AI. It does so by taking inspiration from philosophy of science, to identify various explanatory virtues, and to assess how various existing methods fare with regards to them.

**Strengths:**

Seeking inspiration from the various approaches in philosophy of science to explanation to address the crowded and unsystematic landscape of explanations in AI is a worthwhile strategy.

**Weaknesses:**

I could not make sense of the paper. It never defines clearly what the problem setting is, failing to specify what the explanandum and the explanans are which we are evaluating, it introduces lots of terminology from a disparate and idiosyncratic selection from the literature without properly explaining the terms, the formalism of the entire paper is briefly introduced in a single paragraph without explanation (l. 152), and most of the actual evaluation is moved to the appendix.

Furthermore, the authors implicitly assume that the role of explanations for the purposes of scientific discovery, such as is the case in IBE, is the same as the role of explanations for the purposes of understanding an ML model. Yet on the face of it those roles are very different, and thus without further argument or explanation my confusion about this paper only grows.

Lastly, most of the paper reads like a long list of brief and informal overviews of various parts of the literature and the concepts used in it. There is no real evaluation of any explanatory methods, instead the authors briefly comment on several methods by using the unclear terminology that they introduced before.

**Questions:**

Here is a very incomplete list of both minor and major issues that I came across.

040: "typically" You’re just giving a single example though, that’s a long leap to being typical…

fn: faithfulness is not mentioned in the paragraph...

061: These are very idiosyncratic pickings from the philosophy of science literature, especially when considering that the topic of interest is explanation. (Deutsch is not even a philosopher, let alone an influential one.)

094: Why follow an unpublished manuscript, instead of the many works out there on MI?

102: "good explanations compress ..." Strong claim with no argument. It's also not clear how it connects to the next paragraph. Also, in this paragraph you claim that explanations aim at knowledge. Later you state that they aim at truth... (The more standard view is that they aim at understanding.)

105: "alternative view": Surely that is a common view these days: there exist lots of works trying to interpret the concepts encoded within neural networks.

136: "Explanatory virtues ... truth": That is an unorthodox view of epistemic virtues. The standard view is precisely that truth-tracking does not suffice to explain all of the virtues operating in science.

Section 3.2: IBE is concerned with a very different task than XAI is. In the former, we are trying to distinguish between different hypotheses about the world, and explanations are a tool for getting there. In the latter, we are trying to understand the mechanism by which a fully known NN produces its output. The relation between these projects is not at all clear to me. (Also, lots of terms appear in this section with very little explanation.)

Figure 1, Table 1 and Table 2: these are all placed in the appendix. It is not acceptable to move significant parts of the paper to an Appendix.

Here is a list of some statements whose meaning in the context they appeared was entirely unclear to me without further explanation:

093: "the scientific method"

119: "understanding the neural network" Making clear what that amounts to, is precisely the challenge of explainable AI...

123: "real entities in the world": as opposed to what? Are nodes in the model real entities, for example?

Notation: This is the crucial paragraph explaining what we are explaining, what an explanation is, what a theory is, etc. Yet all we are given is a few symbols with very little context. What are "possible" observations? What is a background theory? Why are explanations "fitted to the data"? (That points to the conflation of explanations in scientific discovery with explanations given that we already know the ground truth theory, namely the NN.)

__Typos:__

043
044
138

---

### Note · Authors · 2025-11-21

I have read and agree with the venue's withdrawal policy on behalf of myself and my co-authors.